# Study on the Spatial and Temporal Evolution of High-Quality Development in Nine Provinces of the Yellow River Basin

Jing Zhang [1], Yang Liu [2], Cuicui Liu [2], Su Guo [2] and Junfu Cui [2,*]

1   Business School, Qingdao University, Qingdao 266071, China
2   School of Economics, Shandong Women's University, Jinan 250300, China
*   Correspondence: 30087@sdwu.edu.cn

**Abstract:** High-quality development (HQD) is a positive initiative in China to promote sustainable development. Promoting HQD in the Yellow River Basin (YRB) is a major national strategy for China. The YRB is an important ecological barrier and economic zone in China, which comprises nine provinces, including Qinghai, Sichuan, Gansu, Ningxia, Inner Mongolia, Shaanxi, Shanxi, Henan and Shandong. The objective of this paper is to provide a comprehensive evaluation of the HQD of the nine provinces in the YRB. It clarifies the basic connotation of HQD and constructs a comprehensive evaluation index system for HQD according to the principles of comprehensiveness, distinctiveness, openness, refinement and feasibility. The comprehensive evaluation index system includes four dimensions: economic development, technology innovation, society and livelihood, and ecological security, which contain 28 secondary indicators. The combined method of coupling development and entropy weight termed the "technique for order preference by similarity to an ideal solution (TOPSIS)" was selected to make a comprehensive evaluation of the HQD of the nine provinces in the YRB from the time dimension and the space dimension, respectively. It was found that there are still problems, such as the differentiation of HQD, the low comprehensive development degree (CDD), and the low development coupling degree (DCD) in some provinces.

**Keywords:** Yellow River Basin; nine provinces; high-quality development; spatial and temporal evolution

## 1. Introduction

Sustainable development has become a goal pursued by countries around the world, including China, which is committed to achieving it. The Yellow River Basin is an important ecological barrier and economic zone in China. As the second largest river in China, the Yellow River is 5464 km long and spans the first, second and third steps in terms of altitude, connecting the Qinghai-Tibet Plateau, the Loess Plateau and the North China Plain as an ecological corridor. It flows through a variety of landscapes, such as deserts, sands and wetlands, as well as several national parks and national ecological function areas, such as Sanjiangyuan and the Qilian Mountains [1]. The YRB has been the political, economic and cultural center of China for more than 3000 years. The *China Statistical Yearbook* shows that the YRB is an important agricultural production area and energy basin in China, accounting for about one-third of the country's grain and meat and more than half of the country's coal reserves. In 2020, the population of the nine provinces in the YRB was 420 million, accounting for 29.8% of the country, and GDP was 2.5 trillion, accounting for 24.9% of the country's income.

While forming a fertile plain in North China, the Yellow River has brought major disasters to the people living along the river. From the perspective of sustainable development, the Yellow River has amongst the highest sediment content, the most difficulties in treatment and the most flood catastrophes of all rivers in the world. From 602 B.C. to 1938 A.D., for about 2540 years, there were more than 1800 breaks and 26 occurrences of large diversions in the Yellow River. Moreover, the frequency of Yellow River breaks has

gradually increased over time [2]. Historically, the main reason why the Yellow River is often broken is the lack of a deep understanding of sustainable development and the need for protection of the ecological environment. The YRB was once well forested, but over the millennia, with increased population, increased production activities and continued irrational development, soil erosion has intensified, and the risk of breaks has greatly increased [3,4]. Regarding the important status of the YRB, there is an old Chinese saying that only when the Yellow River is at peace can the country be at peace [5]. In order to achieve sustainable development of the YRB, in 2019, China proposed to promote high-quality development of the YRB. In 2021, China issued the *Outline of the Plan for Ecological Protection and High-Quality Development of the Yellow River Basin*, which serves as the basis for formulating and implementing relevant planning programs, policy measures and building relevant engineering projects. In the Chinese development process, there has been an east-west gap and a north-south gap in terms of geography, and an adjustment in economic construction, social livelihood, and ecological protection in terms of development priorities [6,7]. So are there similar problems in the development of the YRB, are there development gaps in the upper, middle and lower reaches, and are economic construction and ecological protection not coordinated? In this paper, we undertake a comprehensive evaluation of the HQD in the YRB in order to discuss these issues thoroughly.

This paper constructs a comprehensive evaluation index system for HQD and measures the HQD of the YRB. The contributions of this paper are mainly the following:

(1) Theoretical contributions: A comprehensive evaluation index system is constructed for HQD based on a detailed review of the relevant literature, which can further deepen understanding of HQD.

(2) Practical contributions: HQD represents a positive exploration of the sustainable development of China. This paper focuses on promoting HQD in the YRB, which is conducive to the sustainable development of China.

(3) Research design contributions: A combination of coupling development and entropy weight TOPSIS is selected to measure the HQD of the YRB. Enhanced scientific model selection and more reasonable weight setting, as well as discussion of time and space dimensions, makes the study more comprehensive.

The remainder of the paper is organized in the following sequence: Literature Review, Research Methodology and Data Sources, Empirical Results and Analysis, Discussion and Conclusions.

## 2. Literature Review

Green economy, circular economy and bioeconomy are common terms used in sustainable development research [8]. High-quality development is a positive initiative of China to promote sustainable development. In 2017, it was proposed that the Chinese economy had shifted from the high-speed growth stage to the HQD stage, following which how to promote HQD has become the fundamental basis for measuring the effect of economic development and formulating policy measures. As an important concept, HQD has triggered extensive discussions, and different literatures illustrate the connotation of HQD from different perspectives.

The first category is HQD and the new development concept. A development concept is the precursor of development action, which can indicate the direction of development, lead development ideas, and promote the development of change. In 2016, China put forward the "Five Development Concepts" represented by the concepts of "innovation, coordination, green, openness and sharing". The introduction of the "Five Development Concepts" indicates a deepening understanding of the laws of economic and social development. Innovation is the first driving force, coordination is the endogenous feature, green is the universal form, openness is the necessary path, and sharing is the fundamental purpose [9]. Zhan and Cui [10] argue that the "Five Development Concepts" should be used as a guide to improve the quality of Chinese economic growth and promote economic transformation and development. According to Li and Ren [11], HQD is the approach in

which the new development concept is fully reflected. There is consistency and synchronization between traditional development and HQD. The measurement of HQD needs to be considered in terms of the "Five Development Concepts". Shi and Li [12] propose that the "Five Development Concepts" represent the general direction of HQD, and the construction of an HQD evaluation index system needs to follow these five dimensions. Xu and Ding [13] argue that, in order to promote HQD, we should conscientiously implement the new development concepts and strive to build an all-round development path, including innovation, coordination, green, openness and sharing. Ou et al. [14] argue that HQD involves economic, social, cultural, ecological and other aspects, and that it is a multidimensional concept.

The second category is HQD and efficiency change. One of the most widely used measures of the quality of economic growth in economic growth theory is total factor productivity [15]. The discussion of total factor productivity originated from the economic growth model constructed by Solow [16,17] to study economic growth in the United States. Chow [18] and Chow and Lin [19] studied the economic growth of China in terms of total factor productivity. In the field of HQD, many scholars have drawn on this idea and pursued corresponding explorations. According to He and Shen [20], the modernized economic system promotes HQD through the medium of total factor productivity. In other words, improving total factor productivity is the key to achieving HQD. Liu et al. [21] argued that total factor productivity was a more comprehensive and objective measure of a country's economic efficiency than single factor productivity and can reflect HQD effectively. Liu and Ling [22] proposed that the basis for achieving HQD was total factor productivity, and that, in the current stage of the transformation of old and new dynamics, promoting HQD was needed to accelerate the transformation of the economic structure and promote the steady improvement in total factor productivity. Xu et al. [23] suggested that HQD begins with promoting total factor productivity, and that replacement of old growth drivers with new ones in manufacturing industry involves replacement of industries with lower total factor productivity by those with higher total factor productivity as the core principle. Li and Dong [24] observed that the use of an inclusive green total factor productivity indicator measurement system could enable effective measurement of the level of HQD in the Chinese economy.

From a variety of perspectives, many studies have focused on sustainable development in other economies around the world, including consideration of the circular economy, energy consumption, human capital, and globalization. Gura et al. [25] argued that the circular economy can cope with declining resources, climate change and environmental challenges. Their study identified factors influencing the circular pathway in Albania by exploring green product consumption patterns and found that product labeling, product recycling, instruction manuals, and ingredient details all had an impact. Ren and Albrecht [26] argued that the adoption and diffusion of circular economy innovations facilitated the transition from a linear economy to a circular economy. Their study found that both directive regulation and technology facilitation tools were important in stimulating the adoption of circular economy innovations of small and medium-sized enterprises. Bogovic and Grdic [27] argued that the transition to a green economy involved all elements of sustainable development, and that the implementation of specific green economic policies could contribute to the sustainable development of the EU. Ali et al. [28] argued that the transition to a green economy had the potential to offset environmental, economic and social risks and improve economic development in Ghana sustainably. Therefore, policymakers should implement effective measures to promote the green economy transition. Energy consumption is a key area of concern for sustainable development. Mohammadi et al. [29] argued that energy consumption was closely related to sustainable development and analyzed the relationship between energy consumption and economic growth in developed and developing countries. Their study found that energy consumption had a positive and significant impact on economic growth in both groups of countries. Thadani and Go [30] researched the integration of clean energy into low-cost housing developments in

sustainable cities in Uganda and Indonesia and found that integrating solar energy into low-cost housing was a viable option. Yang et al. [31] focused on the impact of energy consumption on economic growth in East African power pool countries and argued that energy contributed to economic growth in general, while non-renewable energy reduced economic growth in the selected countries. Martínez et al. [32] argued that the sustainable development goals contributed to improving the quality of life of the population, gradually changing the way countries developed and used different technologies. Their study assessed the compliance of Ecuadorian power generation companies with the sustainable development goals. Mpofu [33] argued that green taxes have become key to achieving the sustainable development goals and that green taxes can provide opportunities for green transition policy reforms in African countries, but that green taxes can increase inequality and raise energy costs.

Several scholars have researched sustainable development from a social development perspective. Pitkänen et al. [34] proposed a better understanding of the circular economy from the point view of social sustainability and developed a conceptual framework to analyze and assess the interlinkages between society, the environment and the economy. They explored the establishment of indicators for evaluating the social sustainability of the circular economy in Finland. Singh [35] argued that developing countries, such as Saudi Arabia, should continuously enhance their human capital through targeted education and training programs in order to achieve sustainable development goals. Usman [36] argued that human capital outflow remains a serious challenge for developing countries, especially in sub-Saharan Africa, and that policy makers should pursue effective measures to reduce human capital outflow from the region. Several studies have discussed the impact of external linkages on sustainable development. Lange et al. [37] explored the role and relevance of international trade in the circular economy and found that the sustainable development goals could be furthered by explicitly incorporating circular economy trade into the UK–Canada agreement. Gasimli et al. [38] found that globalization was a determinant of sustainable development in CIS countries, and that CIS countries should be encouraged to integrate into the world economically and politically in order to achieve sustainable development. Wang et al. [39] discussed the green growth of Korea's manufacturing industry and argued that inward FDI had contributed strongly to the green growth of Korea's manufacturing industry. Dornean et al. [40] found that, in EU countries, foreign investors focused on sustainable development environments and that countries with lower carbon emissions were more attractive to foreign investors. Siedschlag et al. [41] focused on sustainability issues in the microsphere, arguing that green innovation was essential to accelerate the transition to a circular economy and that environmental regulations, internal research and development (R&D), and the acquisition of capital assets were important factors that enabled firms to introduce green innovation. Bhagat et al. [42] discussed the impact of Industry 4.0 technologies and green practices on improving the sustainable performance of companies and found that big data analytics had a positive impact on corporate green practices and that the Internet of Things could significantly enhance the implementation of green practices and improve performance.

China promotes sustainable development through HQD. Existing studies have not reached a unified consensus on HQD, but a consistent basic direction has emerged. First, HQD has rich connotations and is a composite concept which needs to completely reflect the new development concept; each development dimension must be included. Second, different dimensions of HQD should be coordinated, a dimension of high development and other dimensions of low development or a dimension of low development and other dimensions of high development are not HQD. Third, HQD has its own specificity. The HQD of different regions and periods are different. These studies provide the foundation for the research in this paper. Studies on sustainable development in other economies around the world also provide useful references for this study.

## 3. Research Methodology and Data Sources

*3.1. The Comprehensive Evaluation Index System of High-Quality Development*

In view of the breadth of HQD, the measurement of HQD requires the establishment of a comprehensive evaluation index system. Compared with a single indicator, the comprehensive evaluation index system has the advantages of including a fuller range of dimensions, reflecting the coordinated development of different dimensions, and considering the distinctiveness of different regions and different periods.

### 3.1.1. Comprehensive Evaluation Ideas at Different Development Stages

Over the past 40 years of reform and opening up, China's evaluation ideas and its indicator system for development have continued to evolve and can be roughly divided into four stages. The first stage, from 1978 to 1993, was the initial stage of opening up to the outside world. In 1978, China began to transfer its focus onto economic construction, and then, in 1987, it was clear that economic construction was the primary task, so the evaluation index system at this stage mainly concentrated on the level of economic development, with economic indicators being given more weight. The second stage was the comprehensive development stage from 1993 to 2007. China put forward major strategies, such as political system reform and building a moderately prosperous society. The development goals and evaluation system in this stage no longer emphasized economic construction alone, but rather, comprehensive development of the economy, culture, politics and society, with more emphasis on people-oriented development.

The third stage, from 2007 to 2015, was the sustainable development stage. In 2007, China proposed achieving good and rapid development of the national economy. That is, China aimed to maintain the stability of economic development, and to promote economic, political, cultural and social construction in a comprehensive manner, adhering to the "five co-ordination" principle. In 2012, China clarified the overall goal of building a fully moderately prosperous society and proposed the overall layout of "five-in-one" economic, political, cultural, social and ecological civilization construction. The evaluation index system of this stage placed more emphasis on sustainability and the synergistic relationship between human and nature, ecological environment and economic growth. The fourth stage was the new development concept stage from 2015 to the present. In 2015, China proposed that, to achieve the development goals of the 13th Five-Year Plan period, the development concepts of innovation, coordination, green, openness and sharing should be firmly established and effectively implemented. In 2017, China clarified that the economy had shifted from the high-speed growth stage into the HQD stage, and the modern economic system should be built. In 2020, China further emphasized that economic and social development in the 14th Five-Year Plan period should firmly implement the new development concept. At this stage, more attention is paid to the quality of development, and the evaluation index system needs to be set more comprehensively and scientifically [43].

### 3.1.2. Principles of Establishing the Comprehensive Evaluation Index System for High-Quality Development

HQD has a connotation of abundance, and a comprehensive index system should be used for accurate evaluation. The construction of the index system should not only closely reflect the connotation of HQD as a development idea to ensure that the evaluation index system is comprehensive and scientific, but also use statistical knowledge as a guide to ensure that the statistical indicators are suitable and operable [9]. The specific principles for the use of indicators are as follows:

The indicators should be comprehensive. The HQD indicator system should be able to reflect the basic features of HQD in all aspects, reflecting the five major development concepts of innovation, coordination, green, openness and sharing, considering both economic growth and social development, and incorporating all indicators, reflecting the connotation and extent of HQD.

The indicators should reflect the characteristics of different regions. As the second largest river in China after the Yangtze River, the Yellow River flows through nine provinces. From the Tibetan plateau to the eastern coast, from the Hehuang culture to the Qilu culture, from the source of three rivers to the delta, each province and each region has different development characteristics due to differences in location conditions, resource endowment, climate environment, etc. The index system should reflect such characteristics according to local conditions [43].

The indicators should be open. Maintaining openness is an important feature of Chinese economic success. Similarly, the construction of an HQD index system should also draw fully on international experience, so that it is based not only on the basic national conditions of China and the basic situation of the YRB, but also has an international perspective.

The indicators should be refined. Simplicity is one of the basic principles of scientific research—under the premise of achieving the same or similar effect, the simpler the theory and method, the more scientific it is. The design of the HQD index system also needs to follow this principle. The index system should include the core indicators—too many indicators will not only increase the difficulty of collecting and processing data, but also dilute the role of the core indicators due to the issue of relevance.

The indicators should be feasible. Real and objective evaluation results should be based on real and objective data, which requires that the data sources for HQD evaluation in the YRB should be real and objective. We should aim to use indicators that are based on directly obtained data and avoid using indicators that involve data projections or alternative scenarios, unless they have reliable theoretical support and can stand up to the test [44].

### 3.1.3. Comprehensive Evaluation Index System of High-Quality Development

With reference to the research results of other scholars, a comprehensive evaluation index system of HQD in the YRB was established according to the development characteristics of nine provinces in the YRB, which included the four dimensions of economic development, technology innovation, social and livelihood, and ecological security, containing 28 secondary indicators, as shown in Table 1.

Economic development is an important aspect of HQD. The economic development dimensions include GDP growth rate, proportion of three industries, foreign trade dependence, general public budget revenue, amount of fixed asset investment, highway route mileage, and freight traffic—all of which are positive indicators. GDP growth rate is used to measure economic growth, and HQD requires economic growth to be in a reasonable range. The proportion of the three industries is used to measure the industrial structure, reflecting the degree of optimization of the industrial structure. Foreign trade dependence is used to measure the external linkage of the economy. The stronger the external linkage, the more foreign economic cooperation. The general public budget revenue is used to measure the financial strength of the government, reflecting the resources available to the government. The amount of fixed asset investment is used to measure capital investment, reflecting the degree of accumulation of local long-term capital. The highway route mileage and freight traffic are used to reflect the infrastructure of the region [45].

The enhancement of technology innovation is the first driving force to promote the HQD of the YRB—we should put innovation at the core of the overall development situation and thoroughly implement the innovation-driven development strategy. The technology innovation dimension includes seven indicators, all of which are positive indicators. They include the R&D expenditure of industrial enterprises above designated size, the full-time equivalent of R&D personnel of industrial enterprises above designated size, the number of full-time teachers in general higher education institutions, the number of Internet broadband access ports, the number of public library collections, the number of patent applications and the technology market transaction value. Technological R&D requires certain basic conditions. The R&D expenditure of industrial enterprises above designated size is used to reflect the financial input. The full-time equivalent of R&D personnel of industrial enterprises above designated size and the number of full-time teachers in general

higher education institutions are used to reflect the human capital situation. The number of Internet broadband access ports and the number of public library collections are used to reflect technological R&D facilities and the reserve of technological R&D materials. The number of patent applications and the technology market transaction value are important indicators of technology output, and the number of patent applications and the technology market transaction value are used to reflect the combination of technology R&D and actual production [46].

**Table 1.** Comprehensive Evaluation Index System of High-Quality Development in the Yellow River Basin.

| Total Index | Primary Indicators | Secondary Indicators | Properties |
|---|---|---|---|
| High-Quality Development | Economic Development | GDP Growth Rate | Positive |
| | | Proportion of Three Industries | Positive |
| | | Foreign Trade Dependence | Positive |
| | | General Public Budget Revenue | Positive |
| | | Amount of Fixed Asset Investment | Positive |
| | | Highway Route Mileage | Positive |
| | | Freight Traffic | Positive |
| | Technology Innovation | R&D Expenditure of Industrial Enterprises above Designated Size | Positive |
| | | Full-Time Equivalent of R&D Personnel of Industrial Enterprises above Designated Size | Positive |
| | | Number of Full-Time Teachers in General Higher Education Institutions | Positive |
| | | Number of Internet Broadband Access Ports | Positive |
| | | Number of Public Library Collections | Positive |
| | | Number of Patent Applications | Positive |
| | | Technology Market Transaction Value | Positive |
| | Society and Livelihood | Per Capita GDP | Positive |
| | | Urban–Rural Income Ratio | Negative |
| | | Number of Health Technicians Per Unit of Population | Positive |
| | | Number of Beds in Medical and Health Institutions | Positive |
| | | Number of Urban Worker Basic Pension Insurance Participants | Positive |
| | | Number of Urban Worker Basic Medical Insurance Participants | Positive |
| | | Urban Registered Unemployment Rate | Negative |
| | Ecological Security | Per Capita Water Resources | Positive |
| | | Per Capita Water Consumption | Negative |
| | | Forest Coverage Rate | Positive |
| | | Afforestation Area | Positive |
| | | Electricity Consumption of 10,000 Yuan GDP | Negative |
| | | General Industrial Solid Waste Disposal Volume | Positive |
| | | Domestic Waste Removal Volume | Positive |

The fundamental goal of HQD is to meet the people's growing need for a better life, which means that the people can gain from HQD. The society and livelihood dimension includes seven indicators: per capita GDP, urban–rural income ratio, number of health technicians per unit of population, number of beds in medical and health institutions, number of urban worker basic pension insurance participants, number of urban worker basic medical insurance participants, and the urban registered unemployment rate, among which, the urban–rural income ratio and urban registered unemployment rate are negative indicators and the rest are positive indicators. Per capita GDP is used to measure people's basic material level, which is the basic indicator to assess the living standard of residents. The urban–rural income ratio is used to measure the income gap between urban and rural residents. The larger the ratio, the larger the gap. The number of health technicians per unit of population and the number of beds in medical and health institutions are used to measure the medical services enjoyed by the residents. The number of urban worker basic pension insurance participants and the number of urban worker basic medical insurance

participants are used to measure the level of social security for residents. The urban registered unemployment rate is used to measure the employment situation, reflecting the number of employment opportunities for residents [47].

The YRB is an important ecological barrier in China, with a fragile ecological environment, severe water scarcity, and limited resource and environmental carrying capacity, so we should promote ecological environmental protection and governance vigorously. The ecological security dimension includes seven indicators: per capita water resources, per capita water consumption, forest coverage rate, afforestation area, electricity consumption of 10,000 Yuan GDP, general industrial solid waste disposal volume and domestic waste removal volume, among which, per capita water consumption and electricity consumption of 10,000 Yuan GDP are negative indicators, and the others are positive indicators. The biggest contradiction in the YRB is the shortage of water resources; per capita water resources is used to measure the water resources available, and per capita water consumption is used to reflect the use of water resources. The most serious problem in the YRB is the fragile ecology; the forest coverage rate is used to reflect the ecological stock, and the afforestation area is used to reflect the ecological restoration. The provinces along the Yellow River are characterized by energy and chemical industry, raw materials, agriculture and animal husbandry, etc. The indicator electricity consumption of 10,000 Yuan GDP is used to reflect the industry's reliance on energy, low quality and low efficiency, while the general industrial solid waste disposal volume is used to reflect quality and efficiency improvement. People's lives are one of the sources of ecological pressure, and the domestic waste removal volume is used to reflect the situation of ecological pressure abatement [48].

### 3.1.4. Coupling Development Model

From the above analysis, it can be concluded that high-quality development requires the consideration of two aspects. On the one hand, it should be stable and developing, referring to the four dimensions of economic development, technology innovation, society and livelihood, and ecological security, which should be rising in level and gradually increasing in strength. Development in quantitative relationships implies a change in quantity between the two periods, where the quantity of a given dimension in the base period is different from the quantity in the reporting period. For positive dimensions, there is a positive relationship between the indicators and HQD. For negative dimensions, there is a negative relationship between the indicators and HQD. Multidimensional development should take a weighted approach to synthesis in order to simplify the discussion [49]. The comprehensive development approach can be portrayed by the following linear model:

$$D_t = \sum_{i=1}^{n} \lambda_{it} \frac{f_{it}}{f_{it-1}} \tag{1}$$

where, $D_t$ is the comprehensive development degree, $f_{it}$ is the $i$ dimension, and $\lambda_{it}$ is the weight of the $i$ dimension. In other words, development is the weighted composite performance of each dimension, and the faster, higher and more effective the development of each dimension $f_{it}$, the faster, higher and more effective the overall development $D_t$.

On the other hand, the quality of development should be high. In addition to the development of each dimension, there should also be synergistic development between each dimension. Some dimensions develop while others do not—some dimensions developing faster while others develop more slowly is not HQD. Synergy draws on the idea of coupling to achieve required outcomes. Coupling is a concept from physics, which is used to describe the situation of system or motion interaction and interdependence—the stronger the system or motion interaction and interdependence, the higher the coupling degree; the weaker the

system or motion interaction and interdependence, the lower the coupling degree. The model for measuring the coupling degree is:

$$C_t = \left[ \frac{\prod\limits_{i=1}^{n} f_{it}}{\left[ \sum\limits_{i=1}^{n} f_{it}/n \right]^n} \right]^{1/n} \tag{2}$$

where, $C_t$ is the development coupling degree. The DCD can be used to measure the synergy of different dimensions; a higher DCD indicates more synergistic development of each dimension, and a lower DCD indicates less synergistic development of each dimension [50].

CDD and DCD combined together can be synthesized into the coupling development degree. The coupling development degree is widely used in the analysis of sustainable development. Qi, et al. [51] studied the interrelationship between citizenship, the regional economy, and public services by means of the coupling development degree in order to promote sustainable development of the population, economy, society, resources and environment. Lu [52] used the coupling development degree to study the coupling development of economy, land, and population in the Yangtze River Delta economic zone.

$$\sqrt{C_t \bullet D_t} = \sqrt{\left[ \frac{\prod\limits_{i=1}^{n} f_{it}}{\left[ \sum\limits_{i=1}^{n} f_{it}/n \right]^n} \right]^{1/n} \sum\limits_{i=1}^{n} \lambda_{it} \frac{f_{it}}{f_{it-1}}} \tag{3}$$

The coupling development degree is the geometric mean of CDD and DCD; the idea of coupling development degree is consistent with the requirements of HQD. First, the coupling development degree can reflect the overall development situation. $D_{nt}$ can accurately measure the CDD of a region in four dimensions: economic development, technology innovation, society and livelihood, and ecological security. Second, the coupling development degree can also reflect the synergy of each dimension, and $C_{nt}$ can effectively measure the synergy of the four dimensions of economic development, technology innovation, society and livelihood, and ecological security. Therefore, the coupling development degree $\sqrt{C_{nt} \bullet D_{nt}}$ can be used to measure $HQD_t$. This paper argues that $HQD_t$ can be defined as the coupling development degree $\sqrt{C_{nt} \bullet D_{nt}}$, which can be used in an exploratory way to measure HQD. That is

$$HQD_t = \sqrt{C_t \bullet D_t} \tag{4}$$

3.1.5. Entropy Weight Technique for Order Preference by Similarity to an Ideal Solution Method

The contribution of different indicators to the whole system is different, so in the comprehensive evaluation, the selection of weights $\lambda_{it}$ is the core of the evaluation, which directly affects the reliability of the evaluation results. The selection of weights should comprehensively consider the degree of independence and variability of indicators and eliminate subjectivity. In this paper, we use the entropy weight technique for order preference by similarity to an ideal solution (TOPSIS) method to determine the weights. This method is a combination of the entropy method and TOPSIS method, which is widely used for weight determination. Hu, et al. [53] used the entropy weight TOPSIS method to determine the weights in their study of the coupling synergy between urban green development and ecological civilization construction in the Yangtze River Delta region. Huang, et al. [54] used the entropy weight TOPSIS method to determine the weights to evaluate the level of water resources management performance in the Yangtze River basin provinces.

The entropy method is an objective assignment method, which can effectively eliminate the subjectivity of assignment. Entropy is also a concept from physics, which is used to measure the degree of uncertainty of a system. A smaller entropy value means less un-

certainty and a larger weight; a larger entropy value means more uncertainty and a smaller weight [55]. The data set according to the evaluation index system of HQD is $\left[x_{ijt}\right]_{m \times n}$; that is, the data of each year is a matrix of $m \times n$, with the rows denoting provinces and the columns denoting indicators, where $m = 9, n = 28$.

$$\left[x_{ijt}\right]_{m \times n} = \begin{bmatrix} x_{11t} & x_{12t} & \cdots & x_{1n-1t} & x_{1nt} \\ x_{21t} & x_{22t} & \cdots & x_{2n-1t} & x_{1nt} \\ \vdots & & \ddots & & \vdots \\ x_{m1t} & x_{m2t} & \cdots & x_{mn-1t} & x_{mnt} \end{bmatrix} \tag{5}$$

Data standardization is based on the idea of the TOPSIS method, which is a multi-attribute decision-making method that measures the superiority of a solution in terms of the distance between the indicator and the "positive ideal solution" and the "negative ideal solution" [56,57]. As a ranking method that approximates the ideal solution, the TOPSIS method has the advantages of easy calculation, a small sample size requirement, and clear and reasonable results. Different standardization methods are used for positive and negative indicators, respectively.

Positive indicators:

$$x_{ijt}' = \frac{\left[x_{ijt} - \min\left(x_{1jt} \ldots x_{mjt}\right)\right]}{\left[x_{ijt} - \min\left(x_{1jt} \ldots x_{mjt}\right)\right] + \left[\max\left(x_{1jt} \ldots x_{mjt}\right) - x_{ijt}\right]} \tag{6}$$

Negative indicators:

$$x_{ijt}' = \frac{\left[\max\left(x_{1jt} \ldots x_{mjt}\right) - x_{ijt}\right]}{\left[x_{ijt} - \min\left(x_{1jt} \ldots x_{mjt}\right)\right] + \left[\max\left(x_{1jt} \ldots x_{mjt}\right) - x_{ijt}\right]} \tag{7}$$

The data is shifted by a fixed constant $\alpha$ to prevent it from being meaningless after taking the logarithm; that is $\left[x_{ijt}''\right]_{m \times n} = \left[x_{ijt}'\right]_{m \times n} + \left[\alpha\right]_{m \times n}$, the probability of the sample $i$ of the indicator $j$ is

$$p_{ijt} = \frac{x_{ijt}''}{\sum\limits_{i=1}^{m} x_{ijt}''} \tag{8}$$

The entropy value of the indicator $j$ is

$$e_{jt} = -\frac{\sum\limits_{i=1}^{m} p_{ijt} \times \ln p_{ijt}}{\ln(m)} \tag{9}$$

The entropy weight is

$$w_{jt} = \frac{1 - e_{jt}}{m - \sum\limits_{j=1}^{m} e_{jt}} \tag{10}$$

The CDD, DCD and HQD all take values in the range of $[0, 1]$. The closer to 0 the worse the HQD situation; the closer to 1 the better the HQD situation, which can be divided into primary HQD $[0, 0.39]$, intermediate HQD $[0.4, 0.69]$, and advanced HQD $[0.7, 1]$ as three major intervals [9].

### 3.2. Data Sources

The data required for the comprehensive evaluation of HQD in the nine provinces of the YRB were obtained from the 2011–2021 *China Statistical Yearbook*, the *China Science and Technology Statistical Yearbook*, the *China Education Yearbook*, the *China Population and Employment Statistical Yearbook*, etc., as well as the statistical yearbooks, science and technology yearbooks, education yearbooks, population and employment statistical yearbooks

of the nine provinces of the YRB. Some data were also obtained from the national and corresponding provincial statistical bulletins, departmental bulletins, etc.

## 4. Empirical Results and Analysis

The evolution of high-quality development in the Yellow River Basin can be analyzed and discussed in terms of two dimensions: the vertical temporal dimension and the horizontal spatial dimension. The time dimension focuses on analyzing the changes in HQD in the YRB from 2010 to 2020, and the spatial dimension focuses on analyzing the changes in HQD in the upper, middle and lower reaches of the YRB. The division of the upper, middle and lower reaches of the YRB was determined according to the hydrological situation and natural environment of the basin, with Qinghai, Sichuan, Gansu and Ningxia as the upper reaches, Inner Mongolia, Shaanxi and Shanxi as the middle reaches, and Henan and Shandong as the lower reaches [58].

### 4.1. High-Quality Development Situation

In terms of time evolution, the level of HQD in the nine provinces of the YRB showed two characteristics. On the one hand, HQD as a whole showed an upward trend. The average value of HQD rose from 0.391 in 2010 to 0.478 in 2020, an increase of 0.086, or 22.0%. On the other hand, HQD showed a divergent trend. The absolute value of Sichuan HQD index increased the most, from 0.548 in 2010 to 0.720 in 2020, an increase of 0.172. Inner Mongolia increased the least, from 0.423 in 2010 to 0.458 in 2020, an increase of 0.035. Qinghai increased the most, from 0.200 in 2010 to 0.277 in 2020, an increase of 39.0%. Inner Mongolia showed the smallest increase, with an increase of 8.2%. Figure 1 shows the HQD of the nine provinces in the YRB at two time points, 2010 and 2020—the HQD of the nine provinces of the YRB has improved significantly, and after 10 years of development, two provinces, Shandong and Sichuan, have entered the advanced HQD zone.

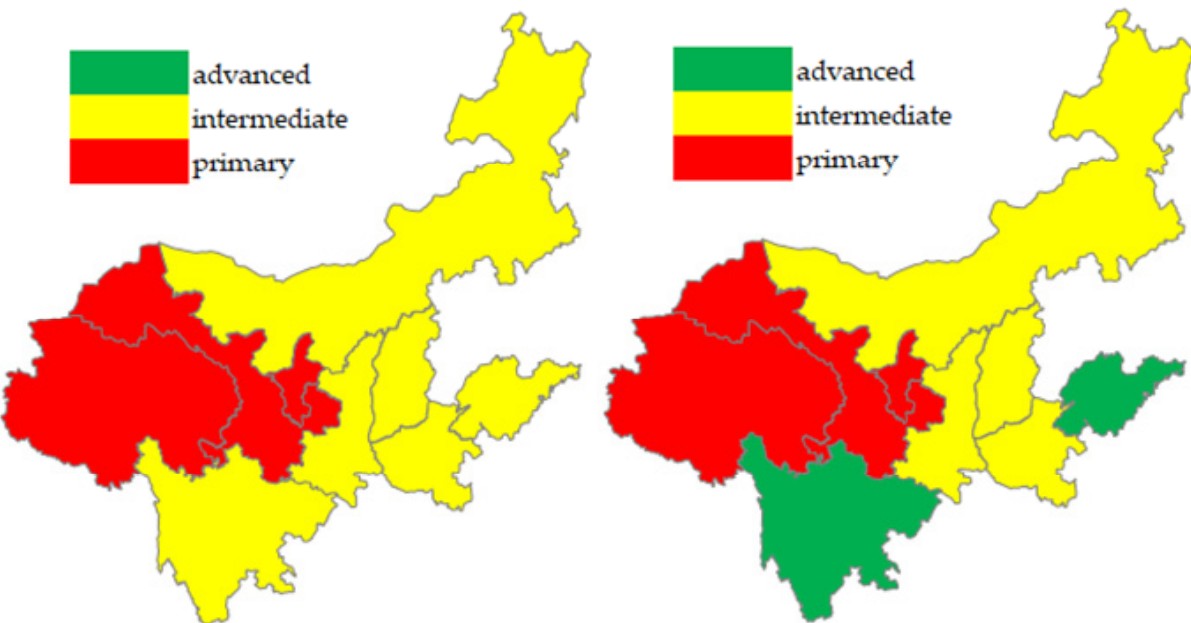

**Figure 1.** High-Quality Development in the Yellow River Basin in 2010 and 2020.

In terms of spatial evolution, there are different trends in the level of HQD in the upper, middle and lower reaches of the YRB. On the one hand, there is a gap between the regional HQD level, showing a pattern of "lower reaches > middle reaches > upper reaches", with the highest level of lower reaches HQD, the lowest level of upper reaches HQD, and the middle level of middle reaches HQD. On the other hand, the gap between the regional HQD levels has been changing. Between 2010 and 2020, the upper reaches and lower reaches HQD indexes improved faster, both by more than 20%, 25.2% for the upper

reaches and 25.0% for the lower reaches. The upper reaches HQD index increased from 0.294 to 0.368, an increase of 0.074, and the lower reaches HQD index increased from 0.583 to 0.729, an increase of 0.146. The middle reaches HQD index improved more slowly, with an increase of 16.2%, from 0.439 to 0.510, an increase of 0.071. After 10 years of development, the gap between the upper reaches and the middle reaches narrowed, and the gap between the middle reaches and the lower reaches widened.

Table 2 and Figures 1 and 2 provide a better analysis of the situation for the vertical dimension and the horizontal dimension of HQD of the nine provinces in the YRB. The high-quality development index is made up of the comprehensive development degree (CDD) and the development coupling degree (DCD). To analyze the trend in HQD comprehensively, we should first analyze CDD and DCD.

**Table 2.** Temporal Evolution of the High-Quality Development in the Yellow River Basin.

| Provinces | 2010 | 2011 | 2012 | 2013 | 2014 | 2015 | 2016 | 2017 | 2018 | 2019 | 2020 |
|---|---|---|---|---|---|---|---|---|---|---|---|
| Qinghai | 0.200 | 0.226 | 0.238 | 0.236 | 0.255 | 0.250 | 0.256 | 0.264 | 0.262 | 0.261 | 0.277 |
| Sichuan | 0.548 | 0.563 | 0.580 | 0.595 | 0.604 | 0.627 | 0.645 | 0.672 | 0.685 | 0.704 | 0.720 |
| Gansu | 0.327 | 0.342 | 0.350 | 0.359 | 0.362 | 0.375 | 0.371 | 0.361 | 0.362 | 0.373 | 0.381 |
| Ningxia | 0.209 | 0.210 | 0.207 | 0.218 | 0.223 | 0.221 | 0.230 | 0.242 | 0.239 | 0.239 | 0.241 |
| Inner Mongolia | 0.423 | 0.434 | 0.448 | 0.462 | 0.445 | 0.452 | 0.463 | 0.464 | 0.461 | 0.470 | 0.458 |
| Shaanxi | 0.461 | 0.471 | 0.480 | 0.508 | 0.513 | 0.528 | 0.526 | 0.545 | 0.562 | 0.582 | 0.588 |
| Shanxi | 0.433 | 0.439 | 0.443 | 0.461 | 0.453 | 0.456 | 0.473 | 0.481 | 0.498 | 0.510 | 0.492 |
| Henan | 0.527 | 0.550 | 0.570 | 0.585 | 0.595 | 0.612 | 0.624 | 0.641 | 0.655 | 0.665 | 0.669 |
| Shandong | 0.646 | 0.671 | 0.681 | 0.707 | 0.694 | 0.727 | 0.731 | 0.751 | 0.766 | 0.776 | 0.795 |
| Average | 0.391 | 0.406 | 0.416 | 0.429 | 0.432 | 0.441 | 0.449 | 0.459 | 0.465 | 0.473 | 0.478 |

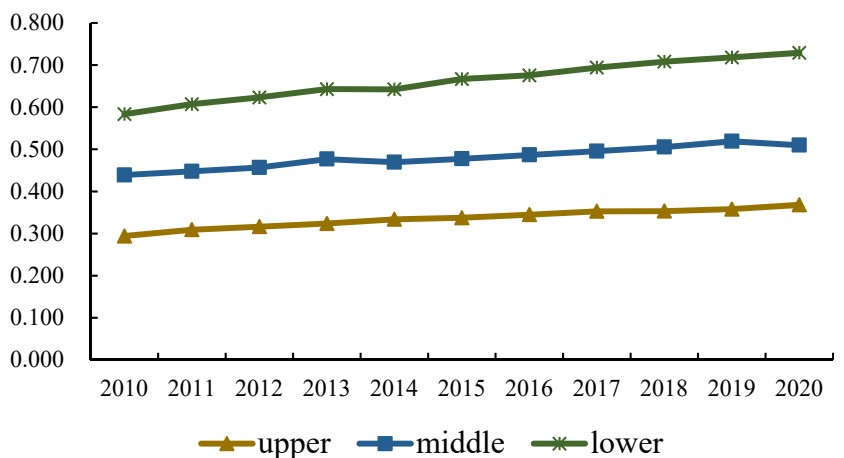

**Figure 2.** Spatial Evolution of High-Quality Development in the Yellow River Basin.

*4.2. Comprehensive Development Degree Situation*

In terms of time evolution, the HQD index of the nine provinces in the YRB was more influenced by the CDD, which showed an overall rising and diverging trend. The average value of the CDD rose from 0.194 in 2010 to 0.259 in 2020, an increase of 0.065, or 33.7%, as shown in Table 3. By province, Shandong had the largest absolute increase, from 0.436 in 2010 to 0.753 in 2020, an increase of 0.317; Ningxia showed a decreasing trend, from 0.065 in 2010 to 0.059 in 2020, a decrease of 0.006; Sichuan had the largest increase, with an increase of 75.8%; and Ningxia had a decrease of 10.2%. Shandong Province, as a lower reaches province, had the highest level of economic and social development among the nine provinces in the YRB, and its economic development, technology innovation, society and livelihood have all grown rapidly. There are many reasons for the rapid development of Shandong Province. First, it possesses obvious location advantages, with convenient transportation. Second, it has better resource endowment, which is suitable for

the development of a variety of types of production. Third, as a coastal province, it opened up earlier and has more institutional dividends compared to inland provinces. Compared with lower reaches provinces, Ningxia, as an upper reaches province, has a large gap in its location advantages, resource endowment and institutional dividends. Although Sichuan is an upper reaches province, it is located in the south of China, and its location advantage, and resource endowment and institutional dividend are also better than the other upper reaches provinces.

**Table 3.** Temporal Evolution of Comprehensive Development Degree in the Yellow River Basin.

| Provinces | 2010 | 2011 | 2012 | 2013 | 2014 | 2015 | 2016 | 2017 | 2018 | 2019 | 2020 |
|---|---|---|---|---|---|---|---|---|---|---|---|
| Qinghai | 0.147 | 0.142 | 0.156 | 0.112 | 0.139 | 0.101 | 0.102 | 0.121 | 0.139 | 0.138 | 0.153 |
| Sichuan | 0.308 | 0.319 | 0.337 | 0.355 | 0.367 | 0.399 | 0.423 | 0.459 | 0.485 | 0.515 | 0.542 |
| Gansu | 0.115 | 0.124 | 0.126 | 0.131 | 0.133 | 0.143 | 0.140 | 0.132 | 0.133 | 0.141 | 0.147 |
| Ningxia | 0.065 | 0.059 | 0.051 | 0.054 | 0.055 | 0.052 | 0.055 | 0.060 | 0.058 | 0.058 | 0.059 |
| Inner Mongolia | 0.206 | 0.214 | 0.235 | 0.242 | 0.210 | 0.216 | 0.226 | 0.234 | 0.228 | 0.245 | 0.220 |
| Shaanxi | 0.225 | 0.232 | 0.238 | 0.264 | 0.269 | 0.286 | 0.284 | 0.305 | 0.327 | 0.358 | 0.372 |
| Shanxi | 0.197 | 0.199 | 0.200 | 0.216 | 0.208 | 0.211 | 0.233 | 0.243 | 0.261 | 0.276 | 0.245 |
| Henan | 0.281 | 0.304 | 0.329 | 0.349 | 0.365 | 0.390 | 0.409 | 0.427 | 0.452 | 0.465 | 0.486 |
| Shandong | 0.436 | 0.477 | 0.498 | 0.546 | 0.542 | 0.593 | 0.608 | 0.636 | 0.666 | 0.672 | 0.753 |
| Average | 0.194 | 0.198 | 0.204 | 0.208 | 0.212 | 0.213 | 0.221 | 0.234 | 0.244 | 0.254 | 0.259 |

In terms of spatial evolution, the HQD of the nine provinces in the YRB was greatly affected by the CDD, as shown in Figure 3. By region, the CDD showed a pattern of "lower reaches > middle reaches > upper reaches", with the highest CDD level in the lower reaches, the lowest CDD level in the upper reaches, and the middle CDD level in the middle reaches. In 2020, the average value of the lower reaches CDD was 0.605, the average value of the upper reaches CDD was 0.163, and the average value of the middle reaches CDD was 0.272. It should be noted that the gap between the upper reaches, middle reaches and lower reaches CDD has gradually widened. Between 2010 and 2020, the lower reaches CDD increased from 0.350 to 0.605, an increase of 0.255, or 72.8%; the upper reaches CDD increased from 0.136 to 0.163, an increase of 0.027, or 20.0%; and the middle reaches CDD increased from 0.209 to 0.272, an increase of 0.063, or 30.1%. The absolute value of the lower reaches CDD improvement was 9.4 times and 4.0 times that of the upper reaches CDD and the middle reaches CDD, respectively, and the improvement rate was 3.6 times and 2.4 times that of the upper reaches CDD and the middle reaches CDD, respectively [59]. In addition to Shandong, Henan, a lower reaches province, has also developed rapidly, with economic development, technology innovation, society and livelihood, and ecological security all showing some degree of improvement, with economic development, technology innovation, and society and livelihood improving the most.

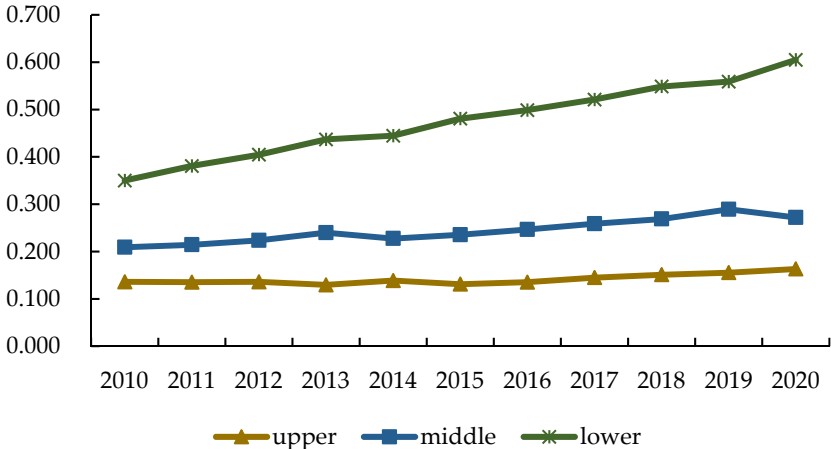

**Figure 3.** Spatial Evolution of the Comprehensive Development Degree in the Yellow River Basin.

### 4.3. Development Coupling Degree Situation

In terms of time evolution, the DCD of the nine provinces in the YRB showed two characteristics. On the one hand, the overall DCD showed an upward trend, as shown in Table 4. The average DCD rose from 0.791 in 2010 to 0.881 in 2020, an increase of 0.090, or 11.4%. On the other hand, the DCD showed a trend of high operation in most provinces and low fluctuation in a few provinces. The DCD of Qinghai was only 0.270 in 2010, which was far below the average, and then gradually increased to reach the highest value of 0.643 in 2016, and then began to decline again to 0.504 in 2020, which was also far below the average. All provinces except Qinghai basically maintained a high DCD, with five of the nine provinces always above 0.9.

**Table 4.** Temporal Evolution of Development Coupling Degree in the Yellow River Basin.

| Provinces | 2010 | 2011 | 2012 | 2013 | 2014 | 2015 | 2016 | 2017 | 2018 | 2019 | 2020 |
|---|---|---|---|---|---|---|---|---|---|---|---|
| Qinghai | 0.270 | 0.358 | 0.364 | 0.498 | 0.468 | 0.619 | 0.643 | 0.575 | 0.496 | 0.497 | 0.504 |
| Sichuan | 0.976 | 0.992 | 0.998 | 0.998 | 0.994 | 0.987 | 0.983 | 0.985 | 0.968 | 0.963 | 0.957 |
| Gansu | 0.929 | 0.942 | 0.967 | 0.979 | 0.982 | 0.983 | 0.988 | 0.990 | 0.986 | 0.981 | 0.990 |
| Ningxia | 0.669 | 0.744 | 0.836 | 0.879 | 0.908 | 0.947 | 0.957 | 0.971 | 0.982 | 0.983 | 0.994 |
| Inner Mongolia | 0.870 | 0.882 | 0.855 | 0.883 | 0.941 | 0.947 | 0.947 | 0.922 | 0.933 | 0.904 | 0.953 |
| Shaanxi | 0.944 | 0.957 | 0.969 | 0.978 | 0.976 | 0.972 | 0.971 | 0.973 | 0.967 | 0.948 | 0.928 |
| Shanxi | 0.953 | 0.968 | 0.979 | 0.985 | 0.990 | 0.985 | 0.963 | 0.954 | 0.950 | 0.943 | 0.987 |
| Henan | 0.988 | 0.994 | 0.990 | 0.979 | 0.970 | 0.960 | 0.952 | 0.963 | 0.949 | 0.951 | 0.919 |
| Shandong | 0.957 | 0.943 | 0.931 | 0.914 | 0.888 | 0.891 | 0.880 | 0.888 | 0.880 | 0.895 | 0.840 |
| Average | 0.791 | 0.832 | 0.845 | 0.883 | 0.883 | 0.913 | 0.914 | 0.903 | 0.885 | 0.880 | 0.881 |

In terms of spatial evolution, the DCD of YRB showed different trends in different regions, as shown in Figure 4. The middle reaches and lower reaches DCD have been running at a high level. From 2010 to 2020, the middle reaches DCD fluctuated between 0.920 and 0.970, and the lower reaches DCD fluctuated between 0.875 and 0.975. The upper reaches DCD showed an overall rising trend, but there was still a gap with the middle reaches and lower reaches. Between 2010 and 2020, the upper reaches DCD rose from 0.636 to 0.830, an increase of 0.194, or 30.5%. In 2020, the DCD gap between the upper reaches and the middle reaches and lower reaches was 0.125 and 0.049, respectively [60]. The DCD of the lower reaches showed a decreasing trend, from 0.972 to 0.879, a decrease of 0.093. The main reason for this trend was that the economic development and technology innovation in Shandong and Henan were improving faster compared to the society and livelihood and ecological security. This also reflects the development characteristics of these provinces, which attach more importance to economic development and technology innovation [61]. The DCD of the upper reaches appeared to be significantly higher, indicating that the dimensions of the upper reaches were more coordinated. However, it is also important to note that the CDD of Qinghai and Gansu has improved more slowly and that Ningxia has even experienced a decline. This coordination presents a certain degree of low-level coordination, which means that the improvement in HQD in the upper reaches resulted more from low-level coordination, indicating that this situation needs to be given sufficient attention [62].

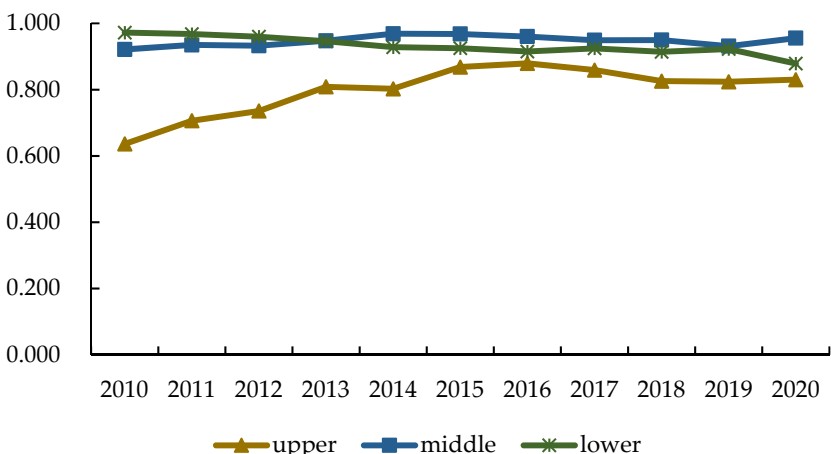

**Figure 4.** Spatial Evolution of Development Coupling Degree in the Yellow River Basin.

*4.4. Robustness Checks*

In this paper, Moran's index (Moran's I), index system adjustment and principal component analysis (PCA) were used to perform robustness tests. The results showed that the measurements in this paper are relatively robust.

4.4.1. Moran's Index

Moran's index (Moran's I) can be used to measure the magnitude of spatial auto-correlation, that is, whether two regions adjacent to each other in geographic space are related or not. The larger the Moran's I, the greater the interspatial interaction and spatial dependence; the smaller the Moran's I, the smaller the interspatial interaction and spatial dependence. The measurement of Moran's I in different periods can reflect the changes in the studied variables in two dimensions: the time dimension and the spatial dimension [63]. Moran's I of HQD, CDD and DCD in the YRB were measured separately, and the results are shown in Table 5. Overall, the Moran's I of HQD in nine provinces of the YRB roughly showed an upward trend, with a certain spatial clustering effect and spatial heterogeneity, which was basically consistent with the discussion of Table 2 and Figure 2. From Moran's I, the spatial agglomeration effect and spatial heterogeneity of the CDD of the nine provinces in the YRB from 2010 to 2020 were higher than the HQD index, showing an overall upward trend, which was basically consistent with the discussion in Table 3 and Figure 3. The Moran's I values of the DCD of the nine provinces in the YRB from 2010 to 2020 were all less than zero, which indicates that there was a spatial dispersion effect in the Yellow River. This is basically consistent with the discussion in Table 4 and Figure 4.

**Table 5.** Moran's I in the Yellow River Basin.

| Indexes | 2010 | 2011 | 2012 | 2013 | 2014 | 2015 | 2016 | 2017 | 2018 | 2019 | 2020 |
|---------|------|------|------|------|------|------|------|------|------|------|------|
| HQD | 0.148 | 0.168 | 0.169 | 0.167 | 0.182 | 0.169 | 0.174 | 0.181 | 0.191 | 0.174 | 0.172 |
| CDD | 0.224 | 0.243 | 0.247 | 0.237 | 0.284 | 0.255 | 0.262 | 0.255 | 0.273 | 0.245 | 0.255 |
| DCD | −0.115 | −0.13 | −0.155 | −0.18 | −0.154 | −0.161 | −0.176 | −0.176 | −0.131 | −0.15 | −0.122 |

4.4.2. Index System Adjustment

In this paper, the comprehensive evaluation index system was adjusted to reduce several indicators at a time to observe the changes in the measurement results. Table 6 shows the results of one of the measurements, with one indicator reduced in each of the four dimensions of economic development, technology innovation, society and livelihood, and ecological security. To be specific, the reduced indicators were the amount of fixed asset investment, the full-time equivalent of R&D personnel of industrial enterprises above designated size, the number of beds in medical and health institutions, and the afforestation

area. The average values of the measured HQD, CDD and DCD were basically the same as the results in Tables 2–4, and the difference was less than 3% in the vast majority of years. The situation of reducing other indicators was basically similar to that shown in Table 6, and the results were relatively stable.

**Table 6.** Index System Adjustment and Gaps.

| Indexes | 2010 | 2011 | 2012 | 2013 | 2014 | 2015 | 2016 | 2017 | 2018 | 2019 | 2020 |
|---|---|---|---|---|---|---|---|---|---|---|---|
| HQD | 0.391 | 0.413 | 0.423 | 0.436 | 0.438 | 0.447 | 0.454 | 0.464 | 0.471 | 0.476 | 0.481 |
| CDD | 0.203 | 0.208 | 0.212 | 0.214 | 0.218 | 0.217 | 0.225 | 0.238 | 0.252 | 0.261 | 0.265 |
| DCD | 0.754 | 0.819 | 0.844 | 0.887 | 0.883 | 0.919 | 0.919 | 0.905 | 0.879 | 0.869 | 0.873 |
| HQD Gaps | 0.1% | 1.7% | 1.7% | 1.6% | 1.5% | 1.3% | 1.2% | 1.1% | 1.2% | 0.6% | 0.7% |
| CDD Gaps | 4.4% | 5.1% | 4.1% | 3.0% | 2.6% | 1.9% | 1.7% | 1.8% | 3.2% | 2.6% | 2.4% |
| DCD Gaps | 4.7% | 1.6% | 0.2% | 0.5% | 0.0% | 0.7% | 0.5% | 0.2% | 0.7% | 1.2% | 0.9% |

### 4.4.3. Principal Component Analysis

PCA is a common method for conducting comprehensive evaluations. This paper used PCA to conduct a comprehensive evaluation of nine provinces in the YRB. The proportion of the first principal component eigenvalue that explained the total variance varied from 56% to 60% in each year, and was around 58% in most years. The first principal component scores of the nine provinces were selected for ranking and compared with the ranking of HQD shown in Table 2. Table 7 shows the situation in 2010, 2013, 2017 and 2020; it can be observed that the rankings of the nine provinces were basically the same, with only a few years slightly different, which confirms the robustness of the method chosen in this paper.

**Table 7.** Principal Component Analysis Ranking and High-Quality Development Ranking.

| Provinces | 2010 | | 2013 | | 2017 | | 2020 | |
|---|---|---|---|---|---|---|---|---|
| | PCA | HQD | PCA | HQD | PCA | HQD | PCA | HQD |
| Qinghai | 9 | 9 | 9 | 8 | 9 | 8 | 9 | 8 |
| Sichuan | 3 | 2 | 3 | 2 | 2 | 2 | 2 | 2 |
| Gansu | 7 | 7 | 7 | 7 | 7 | 7 | 7 | 7 |
| Ningxia | 8 | 8 | 8 | 9 | 8 | 9 | 8 | 9 |
| Inner Mongolia | 6 | 6 | 6 | 5 | 6 | 6 | 6 | 6 |
| Shaanxi | 4 | 4 | 4 | 4 | 4 | 4 | 4 | 4 |
| Shanxi | 5 | 5 | 5 | 6 | 5 | 5 | 5 | 5 |
| Henan | 2 | 3 | 2 | 3 | 3 | 3 | 3 | 3 |
| Shandong | 1 | 1 | 1 | 1 | 1 | 1 | 1 | 1 |

## 5. Discussion

Many countries around the world have made different efforts to achieve sustainable development [25–42], and high-quality development represents active exploration of sustainable development in China. As the Yellow River Basin is an important ecological barrier and economic zone in China, this paper is devoted to the study of HQD in the YRB. In terms of theory, the contribution of this paper mainly lies in exploring the establishment of a comprehensive evaluation index system for HQD to further deepen understanding of HQD. HQD is a complex system project, which needs to consider all aspects of the economy, society, ecology, etc. This paper insists on the principles of comprehensiveness, distinctiveness, openness, refinement and feasibility, and establishes a comprehensive evaluation index system to reflect HQD in four dimensions: economic development, technology innovation, society and livelihood, and ecological security. Based on China's overall HQD [9,43], this study focuses on the HQD of the YRB, considers the development practice of the YRB, and constructs a comprehensive evaluation index system that can reflect the characteristics of the YRB.

In practice, the YRB plays a significant role in China's economic and social development, and promoting the HQD of the YRB is conducive to encouraging China to achieve sustainable development. Studying the HQD of the Yellow River helps to accurately grasp the development characteristics of the region and identify the problems of the region's development in order to pursue more effective measures to promote future development. Based on the comprehensive evaluation index system for HQD of the YRB, this paper quantitatively evaluates the HQD of the YRB and empirically extends the existing research [12,14].

In terms of research design, to make the study more comprehensive and objective, this paper fully draws on the scientific aspects of existing comprehensive evaluation methods and chooses a combination of coupling development and entropy weight TOPSIS, and conducts research in both vertical and horizontal dimensions [51–54]. Overall, relative to existing studies, this paper expands the research related to HQD in terms of connotation interpretation, index system construction, and method selection, and draws some useful conclusions about HQD in the YRB [9,44,45].

Based on the findings, we propose to put forward the following policy recommendations in order to effectively promote the HQD of the YRB and, thus, achieve sustainable development in China.

First, promote cooperation among provinces. The upper reaches, middle reaches and lower reaches of the YRB are an organic whole, and it is important to establish a basin-wide concept, adopt a holistic approach and bring into play collective advantages. The faster developing provinces should drive the slower developing provinces, and the slower developing provinces should take the initiative to learn from the faster developing provinces to form a good trend of the more advanced bringing along the less advanced. Provinces should coordinate planning, improve the policy system, do a good job of policy coordination and convergence, eliminate institutional barriers of resource flow, build a policy highland for common development, and attract resources from all sides to seek common development.

Second, promote development in all dimensions. Promote stable and rapid economic growth, make the economic plate bigger and stronger; adhere to the innovation-driven strategy, put innovation at the core of the overall development, improve the efficiency of resource allocation with innovation, and continuously enhance the contribution of scientific and technological progress; deepen the reform of income distribution, improve the level of public services, make up for the shortcomings of people's livelihood development, and, thus, protect and improve people's livelihoods; promote green development, strengthen the development idea of water as the core, strengthen ecological and environmental management and protection in the YRB, realize the harmonious coexistence of human and environment, and focus on strengthening governance to achieve long-term peace and stability of the Yellow River.

Third, achieve synergy of all dimensions. HQD requires the synergistic development of all dimensions; if some dimensions develop while others do not, or some dimensions develop faster while others develop slower, this is not HQD. This requires adherence to systematic planning, scientific development planning, and anticipation of the development of each dimension. In the process of development, we should take a balanced approach and always pay attention to the synergistic development of each subsystem to ensure progress, while maintaining unity and consistency of pace. In the assessment and evaluation, we should abandon the assessment of a certain dimension, such as the economic dimension, and consider the development of innovation-driven, ecological security and other dimensions comprehensively.

The limitations of this paper mainly arise from the data collection process. On the one hand, it narrows the construction of the comprehensive evaluation index system of HQD in the YRB. HQD has a rich connotation, and the index system constructed in this paper is based on the currently available data. On the other hand, it limits the level of discussion of HQD in the YRB. In addition to macro HQD, HQD also includes meso HQD and micro HQD. Due to data limitations, this study uses macro data extensively and focuses on macro

HQD. For future studies, we will try our best to collect richer and more comprehensive data to improve and deepen the research related to HQD.

## 6. Conclusions

This paper measures the high-quality development of nine provinces in the Yellow River Basin based on the comprehensive evaluation index system and draws the following conclusions [64]:

First, the overall development of HQD is rising, but there is a trend of differentiation. The overall trend of HQD in the nine provinces of the YRB shows an upward trend, and the average value of the HQD index has increased by 0.086 from 2010 to 2020, increasing by 22%; all nine provinces show an upward trend in the HQD index, with an absolute increase between 0.030 and 0.175, with growth of 8.0% to 39.5%. On the other hand, the HQD of the nine provinces in the YRB shows a divergent trend, showing a pattern of "lower reaches > middle reaches > upper reaches", with the HQD of the upper reaches and lower reaches regions improving faster than the middle reaches, and the gap between the upper reaches and middle reaches narrowing, while the gap between the middle reaches and lower reaches is widening.

Second, the comprehensive development is steadily improving, but the level is still not high. As an important part of HQD, the average value of comprehensive development degree as a whole shows an upward trend. From 2010 to 2020, the average value of CDD has improved by 0.065, or 33.7%. On the other hand, one of the main problems that most, if not all, provinces in the YRB are currently facing is the low level of comprehensive development and differentiation. For horizontal comparison, Shandong's comprehensive development level is higher than other provinces. Qinghai, Gansu, Ningxia, Inner Mongolia, Shanxi and other provinces are relatively backward compared with Shandong. Nationwide, there is still a gap between YRB provinces and other Chinese provinces, especially the southern provinces.

Third, the overall development coupling degree is high, but some provinces fluctuate at low levels. Between 2010 and 2020, the average DCD of the nine provinces in the YRB increased by 0.090, or 11.4%. Most provinces in the Yellow River Basin have maintained a high level of DCD, with five of the nine provinces always above 0.9. However, it can also be observed that some provinces, such as Qinghai, still have a low DCD, even below 0.3 in some years, which is not only detrimental to the current HQD, but will also affect the long-term HQD. In addition, the DCD of the lower reaches shows a decreasing trend, indicating that the developments of dimensions are less coordinated. The DCD of the upper reaches shows a significant increase, indicating that the development of each dimension is more coordinated; however, this coordination presents a certain degree of low-level coordination characteristics. These situations need to receive sufficient attention.

**Author Contributions:** J.Z.: conceptualization, methodology, data curation, writing—original draft preparation; Y.L.: software, validation; C.L.: data curation; S.G.: investigation; J.C.: writing—review and editing. All authors have read and agreed to the published version of the manuscript.

**Funding:** This research was funded by the National Natural Science Foundation of China (No.71273148) and Shandong Women's University High-Level Talent Fund (No. 2018RCYJ03).

**Institutional Review Board Statement:** Not applicable.

**Informed Consent Statement:** Not applicable.

**Data Availability Statement:** The data involved in this study are all from public data.

**Acknowledgments:** The authors would like to thank the editors and the three anonymous reviewers. Their constructive comments and suggestions have been very helpful for us to improve this article.

**Conflicts of Interest:** The authors declare no conflict of interest.

## Abbreviations

| Phrase | Abbreviation |
|---|---|
| High-Quality Development | HQD |
| Yellow River Basin | YRB |
| Technique for Order Preference by Similarity to an Ideal Solution | TOPSIS |
| Comprehensive Development Degree | CDD |
| Development Coupling Degree | DCD |
| Research and Development | R&D |
| Moran's Index | Moran's I |

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
