# Peer review of "Study on the Spatial and Temporal Evolution of High-Quality Development in Nine Provinces of the Yellow River Basin"

_sustainability, doi:10.3390/su15086975_

Round 1

Reviewer 1 Report

This paper investigates the economic, social, and environmental development conditions of nine provinces of the yellow river basin in China. The authors introduced the theoretical background of “high quality development” and evaluated how it fits into the nine provinces’ context. Using empirical data from several well-recognized data sources in China, the authors find differences in the fulfilment of “high quality development” across nine provinces and a lack of comprehensive development and low coupling degree in certain areas. Overall, the research topic is important and suitable for the Sustainability journal; the contribution of the paper is clear. I see publication potential in the paper.

A few issues need to be addressed to improve the quality of paper:

First, the theoretical background and economic rationales behind models 1-10 are missing. Please discuss how these models are created and why they are suitable for the analyses. Please also use different measurements as robustness checks. The validity of economic models is the foundation of empirical analyses, please explain how these models are applied in extant literature.

Second, the authors should interpret results after empirical analyses. Please discuss why the results are important, how the results come to existence, and the policy implications for the empirical analyses.

Third, the structure of the paper needs to be reorganized. Please consider using the following structure: 1. Introduction 2. Institutional Background 3. Literature Review 4. Research Methodology 5. Data 6. Empirical Results 7. Discussion 8. Conclusion

Finally, the authors should avoid using a decisive tone and propaganda expressions in an academic paper. For example, “we must promote the high-quality development...”, “the development concept of innovation… must be firmly established and effectively implemented.”, “…openness and sharing must be firmly established..”, and “High-quality development has a rich connotation, and a comprehensive index system must be used for accurate evaluation”. Please simply discuss the importance of the research topic in the institutional background section; don’t oversell the topic—it decreases the research quality. Please also consider using a copyeditor before the resubmission.

Author Response

Dear Anonymous Reviewer,

We do appreciate for your valuable comments and suggestions. In the following, we will respond to each of your concerns and recommendations.

First, the theoretical background and economic rationales behind models 1-10 are missing. Please discuss how these models are created and why they are suitable for the analyses. Please also use different measurements as robustness checks. The validity of economic models is the foundation of empirical analyses, please explain how these models are applied in extant literature.

Response:As suggested by the reviewer, we have revised the Research Methodology section to focus on the merits of the chosen method in this paper and its application in the existing literature. In order to test the robustness of the method proposed in this paper, Moran’s index (Moran’s â… ), index system adjustment and principal component analysis (PCA) are used to perform robustness tests, and the results show that the measurements in this paper are relatively robust.

Second, the authors should interpret results after empirical analyses. Please discuss why the results are important, how the results come to existence, and the policy implications for the empirical analyses.

Response:As suggested by the reviewer, we have revised the Empirical Results and Analysis section to analyze the trends of HQD, CDD and DCD, and discuss in detail the measurement results for some regions and some provinces. In the Discussion section of the paper, policy recommendations are given.

Third, the structure of the paper needs to be reorganized. Please consider using the following structure: 1. Introduction 2. Institutional Background 3. Literature Review 4. Research Methodology 5. Data 6. Empirical Results 7. Discussion 8. Conclusion

Response:As suggested by the reviewer, we have restructured the paper and the current structure of the paper is: 1. Introduction 2. Literature Review 3. Research Methodology and Data Sources 4. Empirical Results and Analysis 5. Discussion 6. Conclusion

Finally, the authors should avoid using a decisive tone and propaganda expressions in an academic paper. For example, “we must promote the high-quality development...”, “the development concept of innovation… must be firmly established and effectively implemented.”, “…openness and sharing must be firmly established..”, and “High-quality development has a rich connotation, and a comprehensive index system must be used for accurate evaluation”. Please simply discuss the importance of the research topic in the institutional background section; don’t oversell the topic—it decreases the research quality. Please also consider using a copyeditor before the resubmission.

Response:As suggested by the reviewer, we have revised the presentation of the paper by eliminating the use of a decisive tone and propaganda expressions. The importance of this research is discussed in the Introduction of the paper (lines 40-63).We also invited other colleagues to spell-check the paper.

All the changes can be found in the revised version (highlight differences between the revised version and the original manuscript).

As authors, we sincerely feel that the quality of paper has been improved after revising it according to the your helpful comments and suggestions. The paper revision process is a very rewarding journey. Once again, we must express our gratitude for your helpful comments and suggestions.

Kind regards,

Authors

Reviewer 2 Report

The paper explores a macro-scale analysis of the high-quality development along the Yellow River Basin. It is a great undertaking and shows nuances in its data collection and analysis methods. To the best of my knowledge, the methodology seems sound. I have some comments that might hinder the legibility of the paper.

Line 30-34 requires citation.

One of the major shortcomings of the paper is its lack of engagement with the international body of literature. The significant majority of the supportive literature is regional. I think addressing the concept of HQD from a more diverse perspective can improve its connection with a broader international audience.

The phrase "high-quality development" (or "high-quality development") has been used repeatedly in the manuscript, which makes some of the sentences difficult to read (for instance, see lines 93–96). I recommend providing abbreviations for frequently used terms (HQD, for instance). You may also provide a list of abbreviations at the end of the paper.

Lines 150–172: Please use the listed paragraph format available in MDPI style. I do not recommend using a full stop after the keyword (e.g. "Distinctiveness.").

Line 178: "containing 28 secondary indicators.", Can these be presented in a table?

Line 179; "Economic development is always the basis and key to solve all problems." This is a very radical sentence that might not be fully supported by the literature, either provide solid citations or rephrase it to make it softer.

The abbreviation R&D appears for the first time on line 197 without any indication of what it means. The first instance of an abbreviation needs to be supported by the full phrase.

TOPSIS has been introduced in the paper with no introduction or citation. What other studies have used the method? What does it stand for? What is its nature? I think TOPSIS needs to be clearly outlined and cited.

Figure 2, Figure 4, and Figure 6: Having a radial graph is not logical since the connection between 2010-2020 is meaningless. A simple chart should be used.

The heading for the "Discussion" should always come before the "Conclusion". A discussion section needs to make a connection with the existing literature and address how the findings of the current study contribute to the existing body of literature. What is more, in the discussion, the plan or suggestions for the future of the HQD in the region could be explored.

I think the idea that the high quality development index (HQDt) can be directly equal to the coupling development degree (Formula 4) needs to be explored before this point. This judgment call seems to be very arbitrary and appears out of nowhere without any support.

Author Response

Dear Anonymous Reviewer,

We do appreciate for your valuable comments and suggestions. In the following, we will respond to each of your concerns and recommendations.

Line 30-34 requires citation.

Response:As suggested by the reviewer, we have added the citation.

One of the major shortcomings of the paper is its lack of engagement with the international body of literature. The significant majority of the supportive literature is regional. I think addressing the concept of HQD from a more diverse perspective can improve its connection with a broader international audience.

Response:As suggested by the reviewer, we have added international literatures [25-42], and discussed sustainable development in other countries and regions.

The phrase "high-quality development" (or "high-quality development") has been used repeatedly in the manuscript, which makes some of the sentences difficult to read (for instance, see lines 93–96). I recommend providing abbreviations for frequently used terms (HQD, for instance). You may also provide a list of abbreviations at the end of the paper.

Response:As suggested by the reviewer, we have povided abbreviations for frequently used terms, and a abbreviations table in the Appendix of the paper.

Lines 150–172: Please use the listed paragraph format available in MDPI style. I do not recommend using a full stop after the keyword (e.g. "Distinctiveness.").

Response:As suggested by the reviewer,we have revised these sentences.

Line 178: "containing 28 secondary indicators.", Can these be presented in a table?

Response:As suggested by the reviewer,we have revised this sentence.

Line 179; "Economic development is always the basis and key to solve all problems." This is a very radical sentence that might not be fully supported by the literature, either provide solid citations or rephrase it to make it softer.

Response:As suggested by the reviewer,we have revised this sentence.

The abbreviation R&D appears for the first time on line 197 without any indication of what it means. The first instance of an abbreviation needs to be supported by the full phrase.

Response:As suggested by the reviewer, the full phrase appears for the first time in line 188.

TOPSIS has been introduced in the paper with no introduction or citation. What other studies have used the method? What does it stand for? What is its nature? I think TOPSIS needs to be clearly outlined and cited.

Response:As suggested by the reviewer, we have revised the Research Methodology section to focus on the merits of the chosen method in this paper and its application in the existing literature.

Figure 2, Figure 4, and Figure 6: Having a radial graph is not logical since the connection between 2010-2020 is meaningless. A simple chart should be used.

Response:As suggested by the reviewer,we have revised these figures.

The heading for the "Discussion" should always come before the "Conclusion". A discussion section needs to make a connection with the existing literature and address how the findings of the current study contribute to the existing body of literature. What is more, in the discussion, the plan or suggestions for the future of the HQD in the region could be explored.

Response:As suggested by the reviewer, we have restructured the paper, and revised the Discussion section, in this part, this paper gives policy recommendations.

I think the idea that the high quality development index (HQDt) can be directly equal to the coupling development degree (Formula 4) needs to be explored before this point. This judgment call seems to be very arbitrary and appears out of nowhere without any support.

Response:As suggested by the reviewer, we have discussed the link between HQD and the coupling development degree, using a more moderate tone in this paper, stating that this is an an experimentation and exploration.

All the changes can be found in the revised version (highlight differences between the revised version and the original manuscript).

As authors, we sincerely feel that the quality of paper has been improved after revising it according to the your helpful comments and suggestions. The paper revision process is a very rewarding journey. Once again, we must express our gratitude for your helpful comments and suggestions.

Kind regards,

Authors

Reviewer 3 Report

The topic of the paper is actual and has social value. The structure of the paper can be improved.

The abstract and the Introduction must be supplemented by the research aims.

Hypotheses are missing.

“Throughout the above discussion” – The use of Discussion is misleading. The ’Discussion’ should be used as a subsection after the results. After this, you have to provide the “Conclusion.”

The Discussion subsection must include references to the scientific literature. This must position the paper, so you have to add in which points you have similar or different results from the previous literature.

Figure3. and 5. need to be elaborated more. There is a lack of comparison. 

The number of scientific sources must be increased by at least 15 sources. All the more so, as there is a lack of international perspective, for example, on other developing countries and their research, such as:

Gura K. S.; Kokthi E.; Kelemen-ErdÅ‘s A. (2021): Circular Pathways Influential Factor in Albania through Green Products Approximation, Acta Polytechnica Hungarica 18 (11) pp. 229–249.

This perspective should appear in the Introduction and in the Discussion too.

A spell check is needed (e.g., ‘developmentin’; ’yellow river’).

A more complex mathematical approach could have been included. 

Author Response

Dear Anonymous Reviewer,

We do appreciate for your valuable comments and suggestions. In the following, we will respond to each of your concerns and recommendations.

Hypotheses are missing.

Response:As suggested by the reviewer, In the Introduction section we have disscussed the importance of this research, and indicates that there are regional development gaps and development focus adjustments in China. This paper proposes the questions “Are there similar problems in the development of the YRB, are there development gaps in the upper, middle and lower reaches, and are economic construction and ecological protection not coordinated?” The research in this paper will answer these questions. These revisions can be found in lines 40-63.

“Throughout the above discussion” – The use of Discussion is misleading. The ’Discussion’ should be used as a subsection after the results. After this, you have to provide the “Conclusion.”

Response:As suggested by the reviewer, we have revised the sentence and restructured the paper.

The Discussion subsection must include references to the scientific literature. This must position the paper, so you have to add in which points you have similar or different results from the previous literature.

Response:As suggested by the reviewer, we have revised the Disscussion section, and shown the contributions made in this paper and added policy recommendations. These revisions can be found in the Disscussion section.

Figure3. and 5. need to be elaborated more. There is a lack of comparison.

Response:As suggested by the reviewer, we have revised Figure3. and 5., and used Moran’s â…  as another method to disscuss the empirical results. These revisions can be found in lines 581-598.

The number of scientific sources must be increased by at least 15 sources. All the more so, as there is a lack of international perspective, for example, on other developing countries and their research, such as:

Gura K. S.; Kokthi E.; Kelemen-ErdÅ‘s A. (2021): Circular Pathways Influential Factor in Albania through Green Products Approximation, Acta Polytechnica Hungarica 18 (11) pp. 229–249.

This perspective should appear in the Introduction and in the Discussion too.

Response:As suggested by the reviewer, we have added international literatures [25-42], and discussed sustainable development in other countries and regions. In the Introduction Section and Discussion Section, we have used these expressions: “Sustainable development has become a goal pursued by countries around the world, and China is committed to achieving sustainable development.”(lines 27-28), “Many countries around the world have made different efforts to achieve sustainable development [25-42], and HQD is an active exploration of sustainable development in China.”(lines 624-626).

A spell check is needed (e.g., ‘developmentin’; ’yellow river’).

Response:As suggested by the reviewer, we have invited other colleagues to spell-check the paper.

A more complex mathematical approach could have been included.

Response:As suggested by the reviewer, we introduced the principal component analysis to test the robustness of the empirical results of the paper. The results of both are basically the same. These revisions can be found in lines 612-622.

All the changes can be found in the revised version (highlight differences between the revised version and the original manuscript).

As authors, we sincerely feel that the quality of paper has been improved after revising it according to the your helpful comments and suggestions. The paper revision process is a very rewarding journey. Once again, we must express our gratitude for your helpful comments and suggestions.

Kind regards,

Authors

Round 2

Reviewer 1 Report

I am happy to see the significant improvements in the current submission. All my prior comments have been addressed properly. I do not have further suggestions. This paper is well-written and will make a good contribution to the literature. 

Author Response

 Dear Anonymous Reviewer,

We are very grateful for your valuable comments and suggestions about our paper, which are highly insightful and enabled us to improve the quality of the manuscript. We sincerely feel that we have learned a lot and this will motivate us to work harder on our research in the future.

Once again, we would like to express our sincere gratitude to you for your constructive comments and suggestions about this manuscript.

Kind regards,

Authors

Reviewer 2 Report

Thank you for your systematic response to the comments. 

Author Response

(The authors gave the same response as above.)

Reviewer 3 Report

The manuscript has been improved a lot. However, I would recommend using a bit fewer abbreviations, as these could improve its understandability.

Author Response

Dear Anonymous Reviewer,

We are very grateful for your valuable comments and suggestions about our paper, which are highly insightful and enabled us to improve the quality of the manuscript. In the following, we will respond to each of your concerns and recommendations.

The manuscript has been improved a lot. However, I would recommend using a bit fewer abbreviations, as these could improve its understandability.

Response:As suggested by the reviewer, we have reduced the abbreviations, mainly in the headings and some of the key sentences.

All the changes can be found in the revised version (highlight differences between the revised version and the original manuscript).

We sincerely feel that we have learned a lot and this will motivate us to work harder on our research in the future.Once again, we would like to express our sincere gratitude to you for your constructive comments and suggestions about this manuscript.

Kind regards,

Authors
